# Mineralocorticoid receptor knockout alters hippocampal CA2 neurons to become like those in CA1
Erin P. Harris[1,4,6], Başak Kandemir [1,6], Stephanie M. Jones[1], Georgia M. Alexander[1], James M. Ward[2], TianYuan Wang[2], Stephanie Proaño[1,5], Xin Xu [3] & Serena M. Dudek [1] ✉

Hippocampal area CA2 has emerged as a functionally and molecularly distinct part of the hippocampus and is necessary for several types of social behavior including social aggression. As part of the unique molecular profile of both mouse and human CA2, the mineralocorticoid receptor (MR; *Nr3c2*) appears to play a critical role in controlling CA2 neuron cellular and synaptic properties. To better understand the fate (or state) of the neurons resulting from MR conditional knockout, we use a spatial transcriptomics approach. We find that without MRs, 'CA2' neurons acquire a CA1-like molecular phenotype. Additionally, we find that neurons in this area appear to have a cell size and density more like that in CA1. These findings support the idea that MRs control CA2's 'state', at least during development, resulting in a CA1-like 'fate'.

The receptors for corticosteroids, glucocorticoid and mineralocorticoid receptors (GRs and MRs, respectively), are important for the regulation of stress responses both in neuronal and non-neuronal tissue[1]. Typically, both receptors can be activated by the circulating stress hormones cortisol or corticosterone and MRs by aldosterone; upon ligand binding, they translocate to the nucleus to operate as transcription factors[2]. Within rodent and human brain, MR (*Nr3c2; NR3C2*) expression is highest in hippocampal area CA2 pyramidal neurons[3–5]. We previously demonstrated that conditional knockout of *Nr3c2* in mice causes a pronounced loss of CA2 marker expression and disrupts CA2's distinct synaptic properties[5]. For example, excitatory synapses in the CA2 *stratum radiatum* are notable for their lack of LTP[6], likely due to the presence of a suite of regulatory molecules that include RGS14, perineuronal nets, and robust calcium handling[7–10]. In addition, CA2-targeted knockout of MR was sufficient to drive some of the behavioral changes reported with MR knockout in excitatory neurons[5,11]. Perhaps related to the finding that mice with a conditional deletion of MR show deficits in social recognition memory[12] is that CA2 has repeatedly been shown to be important for social memory and for a number of social behaviors including aggression and response to social defeat[13–15]. Thus, determining how MRs regulate gene expression specifically in CA2 could reveal insights into the causes of social cognition deficits in a syndromic autism associated with variants in the *NR3C2* gene[16,17].

These neurons in the CA2 region of the MR knockout mice do not die, but what they become remains unknown. Prior studies using whole hippocampal isolates have uncovered much of the role of MRs in regulating gene expression, but as yet, they have been unable to resolve CA2-specific functions due largely to CA2 being a relatively small portion of the total neurons[2,18–20]. Successful approaches such as single-cell RNA sequencing[21] are unlikely to be feasible for CA2-targeted analyses in MR knockout animals because of the loss of many known CA2 cell-type specific gene markers. To begin to bridge this gap, we therefore leveraged the advantages of spatial transcriptomic technology to conduct differential gene expression analysis across hippocampal subfields CA1, CA2, CA3, and the dentate gyrus (DG) of mice with MR conditionally deleted from forebrain neurons (Emx1-Cre; MR fl/fl; 'MR KO'; Supplementary Data Fig. 1a). In addition, we analyzed nuclear density in the CA subregions. Together, our findings indicate that neurons in the area that would have developed into CA2 takes on a more CA1like molecular and cellular phenotype.

## Results and Discussion

As evident by expression of several 'CA2 genes' like *Amigo2*, *Necab2*, and *Pcp4*, the characteristic molecular profile of CA2 neurons is lost in the MR KO mice (Fig. 1a). In addition, genes like *Prkcb* that are low in CA2 of the wild-type (cre-negative; WT) mice, are expressed at levels similar to that in

[1]Neurobiology Laboratory, Division of Intramural Research, National Institute of Environmental Health Sciences, National Institutes of Health, Research Triangle Park, NC, 27709, USA. [2]Integrative Bioinformatics Support Group, Division of Intramural Research, National Institute of Environmental Health Sciences, National Institutes of Health, Research Triangle Park, NC, 27709, USA. [3]Epigenetics and Stem Cell Biology Laboratory, Division of Intramural Research, National Institute of Environmental Health Sciences, National Institutes of Health, Research Triangle Park, NC, 27709, USA. [4]Present address: Neuroscience Institute, Georgia State University, Atlanta, GA, 30303, USA. [5]Present address: Cell Microsystems, Durham, NC, 27713, USA. [6]These authors contributed equally: Erin P. Harris, Başak Kandemir. ✉e-mail: dudek@niehs.nih.gov

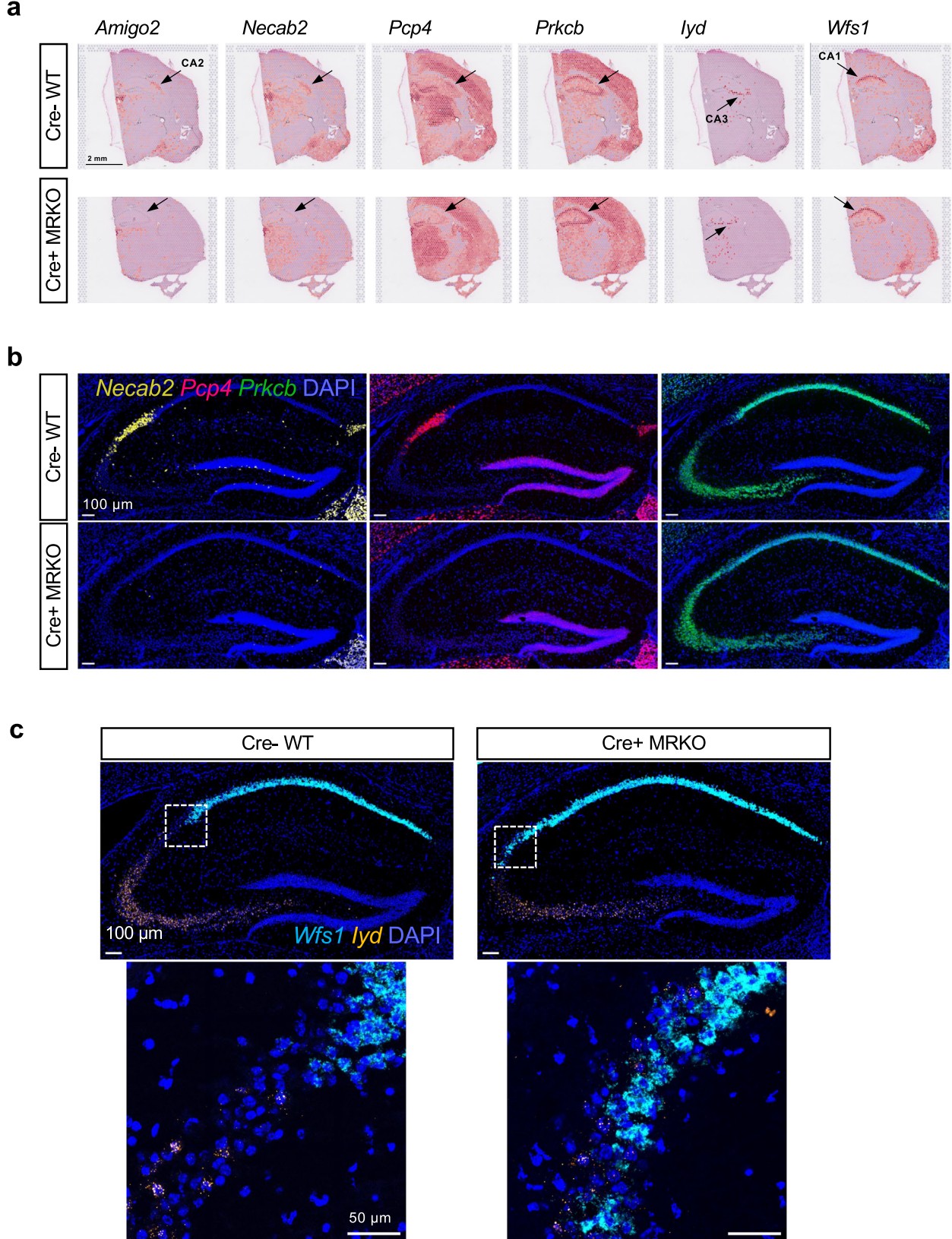

**Fig. 1 | Changes in individual genes with conditional MR KO are evident in spatial transcriptomics and single neurons. a** Examples of brains from a Cre- WT and a Cre+ MR KO mice showing expression of individual genes displayed using the Loupe Browser (10x Genomics). Shown are 'CA2 genes' *Amigo2*, *Necab2*, and *Pcp4*, as well as a 'low in CA2' gene *Prkcb*. Also shown are the 'CA1 gene' and 'CA3 gene' *Wfs1* and *Iyd* respectively. Data are not normalized to spot read counts in these displays. Orange/red = higher expression. **b** Images showing gene expression using single molecule fluorescence in situ hybridization (smFiSH) to validate changes in gene expression with MR KO. Shown are *Necab2*, *Pcp4*, and *Prkcb*. Note the loss of *Necab2* and *Pcp4*, and the filling in with *Prkcb* in the area presumed to be CA2 of the MR KO. **c** smFISH showing that single neurons express neither *Wfs1* nor *Iyd* in CA2 of WT mice. However, neurons in the presumed CA2 region in the MR KO express the CA1 gene *Wfs1*, but not the CA3 gene *Iyd*, with rare examples of neurons expressing both.

CA1 and CA3 in the MR KO mice (Fig. 1a, b). That a 'CA1 gene', Wolfram syndrome-1 *(Wfs1)*, but not a 'CA3 gene', iodotyrosine deiodinase *(Iyd)*, appeared to be expressed in the area that should have been CA2 is strongly suggestive of the CA2 neurons acquiring a CA1-like profile. However, the resolution of the Visium spatial transcriptomics platform is not sufficient to resolve single neurons, so we assessed some of these genes' expression patterns using single molecule RNA fluorescence in situ hybridization (smFISH; Fig. 1b, c). In MR KO tissue, we found evidence of *Wfs1* mRNA in neurons adjacent to *Iyd*-expressing neurons, whereas in WT tissue, a gap of approximately two-hundred microns is present between *Wfs1-* and *Iyd*-expressing cells, with the gap reflecting CA2 neurons. In a few cells, we noted expression of both mRNAs closest to the presumed CA3 border, but largely, the expression of *Wfs1* and *Iyd* was mutually exclusive.

These findings again hint at the idea that neurons that would have become CA2 neurons come to resemble CA1 neurons, rather than CA3 neurons, in the MR KO mice. However, the ability to make such a conclusion requires the advantages of whole genome transcriptomics, so we sought to assess the entire molecular profile of the area CA2 in WT and MR KO tissue. In this case, we manually selected by location the hippocampal subregions for analysis, as automated clustering based on gene expression profiles would necessarily be altered with MR KO (Fig. 2a, Supplementary Data Fig. 2). Here we used data obtained from 3 WT male mice (2 EMX1-Cre-; MR fl/fl and 1 C57BL/6J) compared with 3 conditional MR KO male mice (Supplementary Data Tables 1–3). Similar results were observed with WT and MR KO female mice (1 each was added in Supplementary Data Fig. 5a). Using a principal component analysis (PCA), we found that, consistent with previous data using laser capture microscopy and RNA-seq[10], the spots in CA2 cluster closer to CA3 than to CA1 in the WT tissue (Fig. 2b). However, in the MR KO tissue, the expression profile in spots located in area CA2 no longer cluster with CA3, but instead cluster close to those from CA1 (Fig. 2b). The correlation between CA1 and CA2 (correlation coefficient of 0.810 in Cre- WT) increased significantly in MR KO (to 0.963; Cre+ MR KO; $p < 0.001$) but was unaffected between CA3 and CA2 with MR KO (0.885 in Cre- vs. 0.890 in Cre+).

We observed similar findings using hierarchical clustering (Fig. 2c, Supplementary Data Fig. 3a). Spots representing CA2 and CA3, but not CA1, could be clustered by genotype (Fig. 2d). This analysis also revealed that the changes seen in MR KO were most likely driven by the combined loss of 'CA2 genes' and gain of 'CA1 genes' with minimal changes to the 'CA3 genes' (Fig. 2d). These data strongly suggest that not only are MRs required for the acquisition and/or maintenance of CA2's molecular profile, but also that neurons in this region default to a CA1-like profile in the absence of MRs. Although glucocorticoid receptors (GRs) are upregulated in the MR KOs[5], previous work using a double MR/GR KOs still show a loss of CA2 markers[22], indicating that GRs are not suppressing CA2 gene expression in the MR KOs.

Insights can be gained by looking at the genes up- and down- regulated with MR knockout in the different hippocampal regions. Interestingly, among the CA regions, CA2 had approximately double the number of up-and down-regulated genes than the CA1 and CA3 regions in MR KOs (Supplementary Data Figs. 4, 5b). Among the genes significantly reduced in CA2 were the expected 'CA2 genes' like *Amigo2*, *Pcp4*, and *Necab2*. Similarly, up-regulated genes in CA2 were some of the 'CA1 genes' like *Wfs1* and *Gap43*. The DG, however, had by far the most genes up- and down-regulated in MR KO tissue. In some cases, genes that were highly expressed in both CA2 and DG like *Adcy1* were downregulated in both regions, suggesting a role for MRs as critical regulators of their expression, whereas some, like *Pcp4*, were unchanged in DG. That said, we found no evidence that the DG was fundamentally altered in its molecular identity distinct from the CA regions.

A pathway analysis of genes changed in CA2 of MR knockout tissue revealed networks such as those in NFAT, CREB, GPCR, and synaptogenesis signaling pathways, in addition to circadian rhythm signaling pathways, which might have been expected given the circadian changes in corticosterone[23] (Supplementary Data Figs. 6a, 7a). Some of the upstream

regulators of MR-regulated genes in CA2 showed disease-related genes such as *Prkaa1*, which has been implicated in major depression[24] (Supplementary Data Figs. 6b, 7b, Supplementary Table 4). Of the significant pathways shared with CA1 and CA3, more in CA1 were shared with those in CA2 (Supplementary Data Fig. 7a). In area CA1, most of the pathways impacted by MR KO showed predicted inactivation (down-regulation), notably cell signaling pathways involving opioid, orexin, NFAT, and cAMP signaling, as well as long-term synaptic potentiation (LTP)(Supplementary Data Fig. 8a, Supplementary Table 5). In contrast, the DG showed more pathways with predicted activation (up-regulation), notably endothelin, 14-3-3 mediation, and LTP (Supplementary Data Fig. 8a, e, Supplementary Table 5). In area CA3, however, fewer pathways were predicted to have directional effect (Supplementary Data Fig. 8c, Supplementary Table 5).

Because neuron function relies heavily on synaptic connections, we also investigated the roles of differentially expressed genes in CA2 using ontology terms reported to play a role in synaptic transmission (as provided by the SynGO consortium[25]; Supplementary Data Fig. 6c). Among the identified genes, 135 ( ~ 28%) were mapped to SynGO-annotated genes. Overall, the analysis of overrepresented synaptic gene ontologies revealed 17 cellular component terms and 18 biological process terms that were significantly enriched at a 1% FDR. In CA2, enrichment was observed in both pre- and post-synaptic components. On the postsynaptic side, the enriched processes include regulation of postsynaptic membrane potential by genes such as *Kctd12* and *Gria3*. In addition, genes encoding proteins that are important for synapse assembly or modification of the postsynaptic actin cytoskeleton were also identified as enriched synaptic processes (Supplementary Data Figs. 6c, 8b, d, f, Supplementary Table 6). On the presynaptic side, genes regulating synaptic vesicle exocytosis such as *Prkcb* and *Dtnbp1* were different in CA2 of the MR KO (Supplementary Data Fig. 6c, Supplementary Table 6). Interestingly, CA2 differed from CA1 and CA3 in that it showed a significant enrichment of GO terms representing presynaptic biological processes. For example, synaptic vesicle docking and fusion, which are required for synaptic vesicle exocytosis, were only impacted by MR KO in CA2. Together, these cellular components and functional pathways further emphasize the involvement of MRs in regulating CA2 proteins, implicating them in both pre- and post-synaptic functions.

To investigate further whether MRs are required for the acquisition of structural features of CA2 neurons, we examined cell body density in the different hippocampal areas of WT and MR KO mice. In WT mice, neurons in CA2 and CA3 are larger than neurons in CA1 and so the cell density, as assessed by DAPI stain, is greatest in CA1. Here we measured distance to nearest neighbor and nuclei per $\mu m^3$ (Fig. 3). In both measures, nuclei in CA2 were significantly different from those in CA1, but not different from CA3, in tissue from WT (Cre-negative) animals. This relationship was altered in the MR KO (Cre-positive) tissue in that both distance to nearest neighbor and nuclear density in CA2 were intermediate between CA3 and CA1 (reduced distance between nuclei and increased nuclear density). Both measures were significantly different between WT and MR KO tissue in CA2, but not in CA1 or CA3 (Fig. 3c, d). We interpret these findings to indicate that CA2 neurons, without MRs, come to partially, but not entirely, resemble CA1 neurons.

Taken together, these findings are indicative of MR-dependent influences on neuron development, present in CA2, that are not present in CA1 and CA3. These findings have important implications regarding the underpinnings of a syndromic autism caused by variations in the *NR3C2* gene[16,17]. Further, as MRs and CA2 proteins can be down-regulated by corticosterone in rodents[5], these findings present an avenue by which a part of the hippocampus that regulates social behavior, CA2, could be fundamentally changed during chronic stress.

## Methods
### Animals
All animal procedures were approved by the NIEHS Animal Care and Use Committee (ASP # 2014-0005 and #01-21) and in accordance with the U.S. National Institutes of Health guidelines for the care and use of animals. Mice

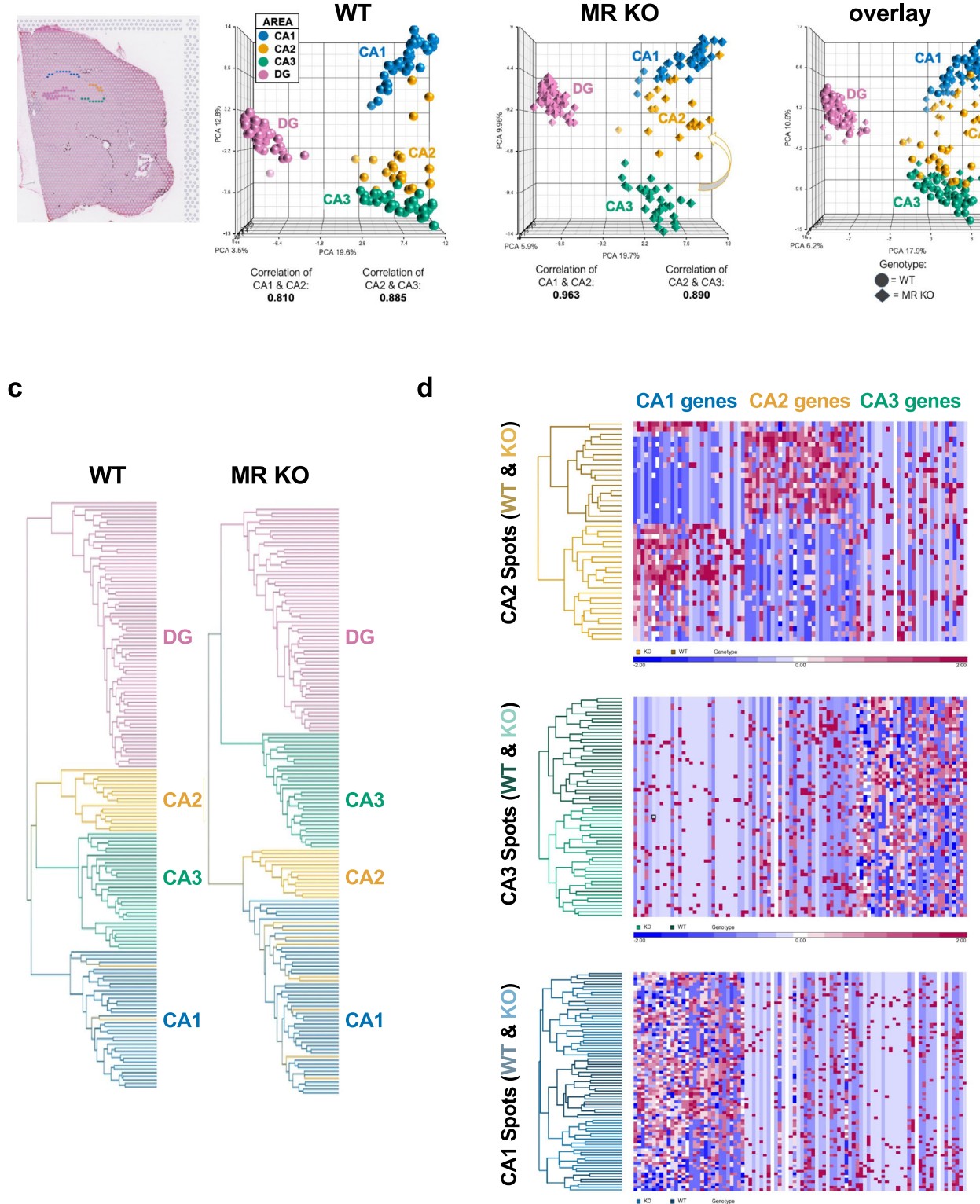

**Fig. 2 | Manual selection of areas allows for analyses of anatomical areas not relying on clustering. a** Location of selected spots in CA1 (blue), CA2 (yellow), CA3 (green), and DG (pink). **b** Principal component analysis (PCA) showing changes occurring with MR KO. Comparison between WT and KO reveals a shift of the transcriptional profile of CA2 in Cre+ MR KO mice towards CA1 as compared with Cre- mice. Correlation between CA1 and CA2 (0.810 in Cre-) increases in MR KO (0.963; Cre+ MR KO) but is unaffected between CA3 and CA2 with MR KO (0.885 in

Cre- vs. 0.890 in Cre+). **c** Dendrograms indicating the hierarchical clustering of spots with similar expression profiles based on 368 hippocampal genes (Supplementary Table 1). Lines representing spots are color-coded according to the spot location. **d** Heatmap of centered, scaled gene expression values for CA2, CA3, or CA1 spots (rows) from WT (top) and MR KO (bottom) males for the top 30 each of CA1 genes, CA2 genes, and CA3 genes (columns) as identified by differential expression analysis (Supplementary Tables 2, 3). Blue = lower expression, Red = higher expression.

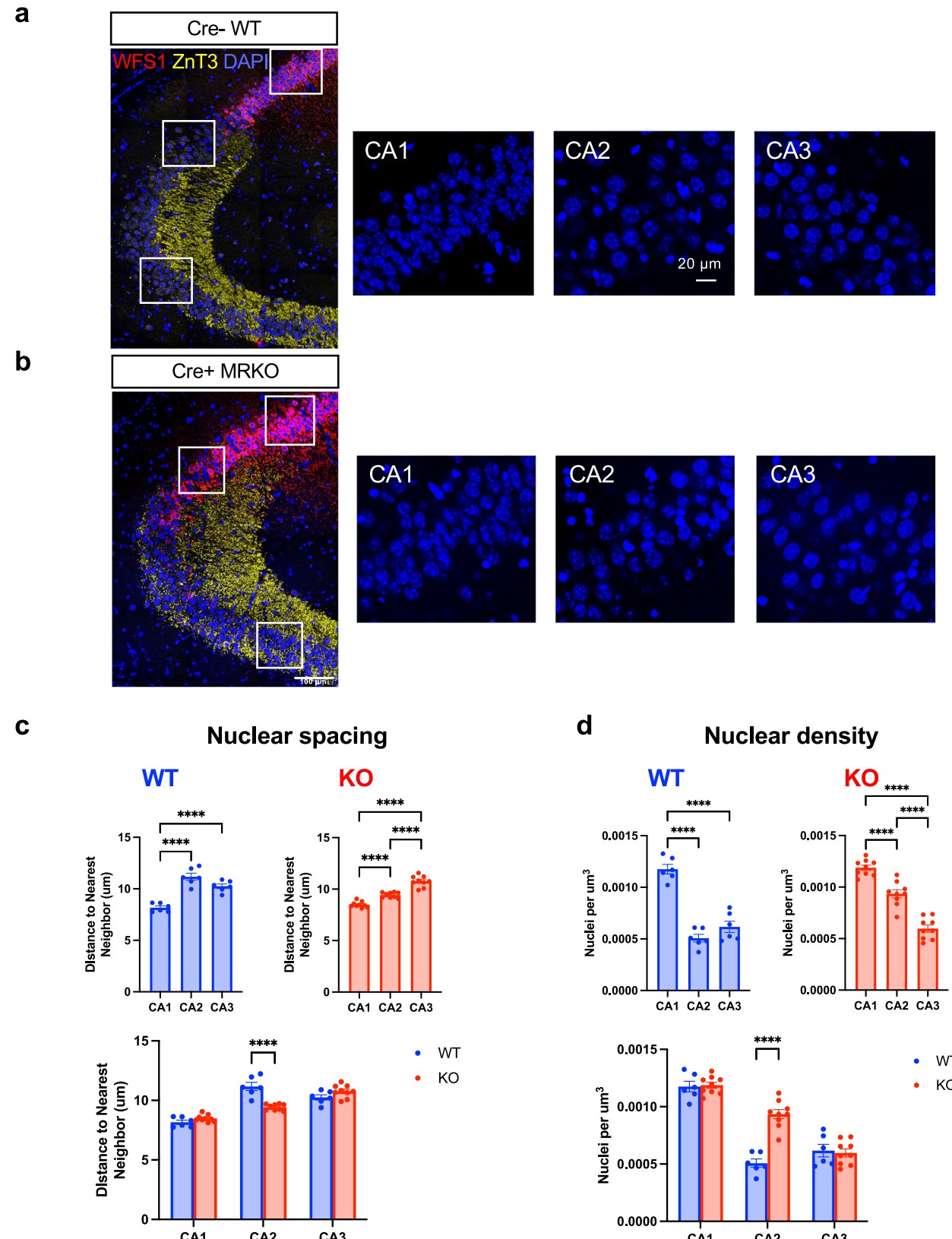

were group-housed under a 12:12 light/dark cycle with access to food and water ad libitum (standard NIH-31 chow from Ziegler Bros, Inc., Gardners, PA). Adult mice, aged 3–4 months, were EMX Cre-positive (heterozygous +/−) or negative (−/−); *Nr3c2* fl/fl (EMX Cre; MR fl/fl[5,26]; EMX Cre: Jackson Laboratory; strain #005628). Mice were backcrossed >10

generations to C57BL/6. One male mouse used in the spatial transcriptomics experiments had no genetic modifications (WT; C57BL/6J). Males were defined as having a Y chromosome. Animals were euthanized with Fatal Plus (sodium pentobarbital, 50 mg/mL; >100 mg/kg) and assessed for lack of response using a toe pinch. Mice were arbitrarily, not randomly, selected for

**Fig. 3 | Pyramidal neuron density in area CA2 becomes more CA1-like in MR KO. a, b** Immunostaining for ZnT3 to label the mossy fibers and WFS1 to label the CA1 pyramidal cells in WT (**a**) and MR KO (**b**) animals were used to localize CA1, CA2 and CA3. DAPI was used to label nuclei. In WT animals, CA2 is positioned at the distal end of the mossy fibers, and WFS1 does not overlap with ZnT3. CA2 was defined as the pyramidal cell nuclei in the pyramidal cell area overlapping with the distal-most 150 μm of the ZnT3 stain, and measurements were made from there. In MR KO animals, ZnT3 expression overlaps with the cells expressing WFS1, so CA2 was defined as the area of overlap between these two markers, and measurements corresponding to CA2 were made from that area. Inset squares represent the areas in each subregion that are shown enlarged on the right side. **c** Nuclear spacing, measured by nearest neighbor analysis, show that in WT animals, both CA2 and CA3 neurons have significantly greater nuclear spacing than CA1, and CA2 spacing does not differ from CA3 ($F_{(2,10)} = 81.24$, $p < 0.0001$, one-way ANOVA; results of Tukey's post-hoc test shown). In MR KOs animals, both CA2 and CA3 have significantly greater nuclear spacing than CA1, but CA2 has significantly less nuclear spacing than CA3. ($F_{(2,16)} = 122.3$, $p < 0.0001$, one-way ANOVA; results of Tukey's post-hoc test shown) Nuclear spacing in CA2 was significantly greater in MR KOs than in WTs (main effect of subfield: $F_{(2,26)} = 164.7$, $p < 0.0001$; subfield x genotype interaction: $F_{(2,26)} = 44.71$, $p < 0.0001$; two way-ANOVA, result of Sidak post-hoc test shown). **d** Nuclear density was measured across the depth of DAPI-stained section. In WT animals, CA2 had a significantly lower nuclear density than CA1 but not CA3 ($F_{(2,10)} = 97.75$, $p < 0.0001$, one-way ANOVA; results of Tukey's post-hoc test shown, but in MR KO animals CA2 nuclear density was significantly greater than in CA3 ($F_{(2,16)} = 146.7$, $p < 0.0001$, one-way ANOVA; results of Tukey's post-hoc test shown). Nuclear density was significantly greater in CA2 of MR KO animals than in WT (main effect of subfield: $F_{(2,26)} = 210.0$, $p < 0.0001$; main effect of genotype: $F_{(1,13)} = 9.863$, $p < 0.001$; subfield x genotype interaction: $F_{(2,26)} = 35.31$, $p < 0.0001$; two way-ANOVA, result of Sidak post-hoc test shown), suggesting that CA2 moves toward an anatomical profile more closely resembling CA1 than CA3.

use. Twenty-five mice were used in this study: spatial transcriptomics, 8 mice; smFISH, 2 mice; and histology, 15 mice.

### Spatial Transcriptomics

Brains from 3 control mice (2 EMX Cre-negative and 1 C57BL/6J; WT) and 3 Cre-positive (MR KO) male mice were removed following rapid decapitation and frozen in isopentane on dry ice. A fourth pair of animals were used for follow-up validation using 1 each Cre-negative and Cre-positive female mice from part of a separate study in order to assess consistency, variability, and applicability across sexes. This replicate was processed using same analysis approach. Tissue was stored at −70 °C until use, at which time they were cryosectioned at 10 μm and mounted on Visium slides (10x Genomics, Cat. #1000184). Slides were processed per the 10x Genomics protocol, as used by Vanrobaeys, et al.[27]: tissue was fixed with methanol and stained with H&E followed by imaging. Optimal permeabilization time determined by a tissue optimization kit (10x Genomics, 1000193) was 18 min. The on-slide cDNA synthesis and release, cDNA amplification, and library preparation for RNAseq were performed according to the manufacturer's protocol. Library samples were sequenced on an Illumina NexGen Sequencer (NovaSeq 6000 with read 1 for 28 nt and read 2 for 90 nt length) with an average read depth of 233 million read pairs per brain section, and 86,260 reads and 3,822 genes per spot. Distributions of hippocampal read counts per spot, number of genes per spot, and percent mitochondrial DNA per spot are shown in Supplementary Data Fig. 1b.

For processing of raw spatial transcriptomics data, raw FASTQ files along with slide image for each sample were processed using Space Ranger software (version 1.1, 10× Genomics) to align reads against mouse reference genome mm10. A feature-spot matrix was generated using the Visium spatial barcodes. Seurat (version 4.0) was used to perform clustering analysis of combined dataset[28]. SCTransform was applied to normalize the gene expression values across each spot[29]. Projection maps were created using gene expression across six brain samples following Seurat with Harmony layer integration[30]. Differentially expressed genes (DEGs) were identified by pair-wise comparisons between the 3 WT and 3 MR KO samples within the manually selected spots in CA1, CA2, CA3 pyramidal cell, and DG granule cell layers. Note that although the location of CA2 could be identified in 10xGenomic's Loupe browser based on gene expression (expression of representative 'CA2 genes') and anatomical landmarks, this was not possible for identification of CA2 in the knockout due to loss of such marker gene expression. Thus, the experimenter was not blinded to genotype. In this case, anatomical location alone was used to select spots representing CA2. A gene was considered expressed if its normalized expression value was greater than 0.2 in at least two out of the three samples within at least one group (genotype). Principal component analysis (PCA) and hierarchical clustering of these spots were generated by the Genomics Suite of Partek software package version 6.6. Correlations were tested for significant differences using Fisher z-transformed correlation values,

followed by a standard test in a normal distribution using pnorm() in R. Volcano plots were constructed using VolcaNoseR[31]. Venn diagrams were generated by Venny 2.1[32]. Differentially expressed genes between Cre- WT and Cre+ MR KO in CA2 with log2 fold-change values greater than 0.585 or less than −0.585 and FDR < 0.05 were included for Ingenuity Pathway Analysis (IPA; Qiagen).

### Single-molecule fluorescent in situ hybridization and imaging

We performed single-molecule fluorescent in situ hybridization (smFISH; RNAscope) to validate gene expression changes detected by spatial transcriptomic sequencing and to assess single-cell distribution of gene transcripts. Two animals, one per genotype (Emx1 Cre + /-: MR fl/fl); were sacrificed via rapid decapitation. Brains were extracted and flash-frozen in Tissue Plus® O.C.T Compound (Fisher Scientific, Hampton, NH) in isopentane chilled over dry ice. Brains were stored at −70 °C until cryosectioned at 20 μm and mounted on SuperFrost® Plus slides (Fisher Scientific). Tissue sections were probed for target mRNAs according to the manufacturer protocol for the RNAscope Fluorescent Multiplex kit (Advanced Cell Diagnostics, Hayward, CA). Target probes for in situ hybridization were Mm-*Nr3c2*-E5E6-C3 (Cat#456321-C3), Mm-*Pcp4*-C2 (Cat#402311-C2), Mm-*Necab2*-C3 (Cat#467381C3), Mm-*Prkcb*-C1 (Cat#874311), Mm-*Wfs1*-C3 (Cat#500871-C3), and Mm-*Iyd*-C2 (Cat#465011-C2) from Advanced Cell Diagnostics (Hayward, CA). Signal was developed using Opal Dyes 520 (Cat#OP001001), 570 (Cat#OP-001003), 690 (Cat#OP-001006) from Akoya Biosciences (Marlborough, MA). Sections from both genotypes were processed in parallel and imaged on a Zeiss LSM 880 inverted confocal microscope. Whole hippocampal images were acquired with a Plan-Apochromat 20×/0.8 M27 objective using z-stacks which were collapsed with a maximum intensity projection. 63× images were acquired with a Plan-Apochromat 63×/1.40 Oil DIC M27 objective with a pinhole setting to yield a Z-thickness of 1.7 μm and capture the coexpression of transcripts within the same cells. Acquisition settings were set separately for each staining scheme and held constant across genotypes. Brightness/contrast adjustments were made in FIJI/ImageJ and Powerpoint and applied equally across genotypes for comparison purposes.

### Histology

Animals from 4 litters (Emx1 Cre + /-: MR fl/fl; 6 WT/Cre- (2 male, 4 female) and 9 MR KO/Cre+ littermates (2 male, 7 female) were transcardially perfused with a phosphate buffered saline (PBS) flush, followed by 4% paraformaldehyde in PBS and brains post-fixed overnight in the same fixative. Brains were sectioned at 40 μm with a vibratome (Leica vt1000) and stored in PBS with sodium azide. For immunohistochemistry, sections were washed in PBS followed by PBS with 0.1% triton X-100 (PBST). Sections were blocked in PBST with 5% normal goat serum for 1 h followed by overnight incubation in blocking solution plus primary antibodies: rabbit anti-WFS1 (1/500, Proteintech, 11558-1-AP) and guinea pig anti-ZnT3 (1/500, Synaptic Systems, 197 004). Sections were washed in PBST then

incubated for 2 h in blocking solution plus secondary antibodies: goat anti-rabbit 568 (1/500, Invitrogen, A11011) and goat-anti guinea pig 633 (1/500, Invitrogen, A21105). Sections were washed with PBST then PBS, mounted and coverslipped with Vectashield hardset mounting medium with DAPI (Vector Laboratories, H-1500). Sections were imaged on a Zeiss LSM 880 confocal microscope at 40X using tiled Z-scans. Imaris (Oxford Intruments) software was used to perform nearest neighbor analysis and measure total volume of analyzed regions to arise at measures of nuclear spacing and nuclear density. Statistical analyses were performed using GraphPad Prism software. The experimenter was blinded to genotype through imaging, but they could not be blinded once the experimenter examined the staining because of the distinct staining patterns in the WT and KO.

### Statistics and Reproducibility
Measurements were taken from distinct brain samples, except for the histology, where measurements were made from both sides of the brain (2 hippocampi within an animal) and were averaged. See individual sections above for statistics, animal numbers, and extent of experimenter blinding.

### Reporting summary
Further information on research design is available in the Nature Portfolio Reporting Summary linked to this article.

### Data availability
The datasets generated and/or analyzed in the current study are available in the Gene Expression Omnibus (GEO), accession number: GSE272919. Original images are available from the corresponding author upon reasonable request.

### Code availability
SynGO is available at https://www.syngoportal.org/. Venny is available at https://bioinfogp.cnb.csic.es/tools/venny/index.html. VolcaNoseR is available at https://huygens.science.uva.nl/VolcaNoseR/.

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

## Acknowledgements

We would like to thank Eli Ney of the NIEHS Comparative & Molecular Pathogenesis Branch for scanning the Visium slides and Victor Catalan Gallegos and Priyanka Singh for cutting the brain tissue. We also thank members of the NIEHS Fluorescence Microscopy and Imaging Center. This research was funded by the Intramural Research Program of the NIEHS, NIH, ES100221.

## Author contributions

S.D., G.A., and S.P. conceived of the study. Experiments were performed by E.H., S.J., G.A., S.P., and X.X. Data were analyzed by E.H., G.A., T.Y.W., J.W., and B.K. S.D., E.H., B.K., S.J., and G.A. wrote the paper with input from the other authors.

## Funding

## Competing interests

The authors declare no competing interests.
