## [Peer Review File · Communications Biology]

Fate (or state) of CA2 neurons in a mineralocorticoid receptor knockout.

Corresponding Author: Dr Serena Dudek

Version 0:

Reviewer comments:

Reviewer #1

(Remarks to the Author)

The findings by Harris et al. indicate that MR-dependent influences on neuron development are present in CA2 but not in CA1 and CA3. These findings have important implications for understanding the underlying mechanisms of syndromic autism caused by variations in the NR3C2 gene. Additionally, since mineralocorticoid receptors (MRs) and CA2 proteins can be downregulated by corticosterone in rodents, this research suggests a pathway by which chronic stress could fundamentally alter CA2—a region of the hippocampus involved in regulating social behavior. The manuscript is well-written, and the results are clearly explained. However, I have a minor concern regarding the sample size, as they used only three biological variants. How relevant are their findings given this limitation?

Reviewer #2

(Remarks to the Author)

Summary

This study follows on from groundbreaking earlier work to characterise the changes occurring in the CA2 area of the hippocampus following conditional knock out of MR, addressing a key gap in current literature and generating important new knowledge. This work may provide new insights relevant for the study of autism, a syndrome linked to MR gene variants.

Overall impression of the work

The manuscript is clearly written and flows well. Methods used are well justified although generally lacking in detail. The outcomes are precise measurements of gene expression analysis to characterise the neuronal identity of this area of the brain, providing spatial resolution of changes. The work contributes new insights into the to the research field . Main conclusions are justified.

Specific comments, with recommendations for addressing each comment

1. Methods should include more detail about animal experiments. The ARRIVE guidelines offer a good framework to adhere to in order to maximise the reproducibility and reuse of the work. Details about the underlying genetic background of all mice used in the different analysis, housing conditions and husbandry should be included. How were mice allocated to each experiment and at what stages were experimenters blinded to genotype? It is not entirely clear if manual spot selection was used for both WT and KO samples, or just for KO samples. Whilst it is understandable why automatic selection was not possible for KO mice, the default should be to apply manual selection based on anatomical location to both sample types by an experimenter blinded to genotype so 'CA2' spot selection was standardised across samples since using different methods could introduce bias. Please provide more detail of how regions (spots) were selected in both types of samples to ensure this is clear. If no blinding/randomisation took place state this explicitly. Whilst studies on female mice have been conducted the data is not presented, why?
2. Please justify n-numbers used for animal studies in text (power calculation including effect size, variance and power or other justification). Also include a statement about the total number of mice used in this manuscript (ST, 6 mice; SMF, 2 mice; Histology, 15 mice = 23?) and detail any ethical approvals for the study (this was ticked in checklist but I could not find this statement in the manuscript).
3. What was the inclusion overlap for the DEG analysis i.e. were all genes expressed in any sample included or did you have a stringency cutoff of gene must be expressed in 2/3 samples per group or all 3/3 samples? What was the threshold for the gene to be expressed?

4. Fig2. Include name of test used for correlation between groups.

5. Extended data 2: More information required relating to this figure. Assignment of cell types to clusters should be provided on the figure or in the legend. Please elaborate on the occurrence of batch effects, I assume from the figure that WT1, 3 and KO1 were analysed separately from WT2, KO2 and KO3. If so, these batch effects seem substantial. In light of such clear batch effects how can the results be relied upon? You state that “Note images of WT3, KO2, and KO3 were flipped horizontally 330 for illustrative purposes.” Is this correct (WT2?) and is it relevant given low-dimensional embedding produced by UMAP are preserved regardless of how the embedding is rotated or reflected? Also there is a 1 missing off the bottom UMAP2 axis (should read 10). Please identify the different wildtype backgrounds identified in Extended Fig. 2A (Nr3c2 fl/fl and 1 C57B6J wildtype).

6. Extended data 5: “Upstream regulators affected by MR KO in CA2 are shown. PRKAA1 and SPRY2 regulators were found to be ‘inactive’ in the regulation of target genes (activation z-score < 0), while other regulators, such as MECP2, were ‘active’ in the regulation of target genes like FKBP5 (activation z-score > 0)”. This interpretation of the upstream regulation doesn’t make sense to me the up stream regulators are not ‘inactive’ the score is worked out on the data from their downstream targets in the data set. The overlap p-value indicates if there is a statistically significant overlap between genes in the dataset and those regulated by the upstream regulator, all of which are highly significant. The z-score indicates the magnitude of the regulation and the sign reflects the predicted activation state, Values <-2 or >+2 are considered significant with a positive number predicting activation of the upstream regulator and a negative value predicting inhibition of the upstream regulator. So based on table b PRKAA1 is predicted to be significantly inhibited as an upstream regulator so its down stream targets will be up or downregulated depending if PRKAA1 affects their expression in a negative or positive way respectively. Whilst the other upstream regulators may be playing a role (due to significant overlap of target genes) the directionality cannot be predicted as z-scores are less than ±2. See Microsoft Word - Upstream Regulator Analysis_Whitepaper_08-22-12_KB.docx (ingenuity.com) for more details.

Reviewer #3

(Remarks to the Author)

This is a short manuscript with extensive data that show that in absence of the mineralocorticoid expression, CA2 pyramidal neurons in the mouse hippocampus do not properly differentiate (as shown earlier), but rather acquire a CA1-like phenotype. This is of relevance, given that absence of MR is associated with autism, and CA2 is important for social memory formation. And, it is, or would be, fascinating to learn about out how this receptor for a stress/circadian hormone affects neuronal end-differentiation.

The authors combined spatial transcriptomics with RNAseq, in situ hybridization, and cell nuclear density analysis to make their point. In addition to their main point, the data also are very informative about the MR-dependent genes in the other hippocampal cell types, something that is overdue – and very good to see.

There are no major issues regarding the approach or main conclusion. Indeed, snRNAseq is not appropriate, given that it would be impossible to trace the ‘former’ CA2 neurons in the mutants. As for additional data – it would of course be interesting to know what happens to the projections to the ‘cells formerly known as CA2 neurons’. But I can see that that is certainly a different project.

Depending on the plans of the authors, it might be nice to see the pathway enrichment analyses for the other subfields. The data also may contain interesting facts in relation to neurogenesis and apoptosis in the DG, but there I can also see that the framing of the story would become less focused. For one thing, the data on differentially expressed genes in the other subfields should become available.

One question which may in fact be resolved with immunohistochemistry (or perhaps was addressed in previous work): what of the heterozygous animals? Is half the gene dose sufficient to have the CA2 develop? That may also be relevant in relation to the ASD that develops as a consequence of MR mutations.

Minor:

I. ‘including’ – which other corticosteroid receptors are there? (I guess perhaps the AR for DHEA, but nomenclature would say MR and GR are all?).

Version 1:

Reviewer comments:

Reviewer #1

(Remarks to the Author)

The authours answered my questions.

Reviewer #2

(Remarks to the Author)

My queries have been sufficiently addressed.

Reviewer #3

(Remarks to the Author)

I am happy with the revision and the authors' explanations. And commend reviewer 2 for their thoroughness.

Important data for understanding of hippocampus function, and understanding of corticosteroid receptors

Response to Reviewers

We thank the reviewers for their helpful suggestions and constructive comments. We have taken their suggestions and accordingly, we have made the following revisions listed below. We appreciate the opportunity to strengthen our manuscript.

Reviewer #1:

The findings by Harris et al. indicate that MR-dependent influences on neuron development are present in CA2 but not in CA1 and CA3. These findings have important implications for understanding the underlying mechanisms of syndromic autism caused by variations in the NR3C2 gene. Additionally, since mineralocorticoid receptors (MRs) and CA2 proteins can be downregulated by corticosterone in rodents, this research suggests a pathway by which chronic stress could fundamentally alter CA2—a region of the hippocampus involved in regulating social behavior. The manuscript is well-written, and the results are clearly explained. However, I have a minor concern regarding the sample size, as they used only three biological variants. How relevant are their findings given this limitation?

We thank the reviewer for the supportive comments!

Regarding sample size for the spatial transcriptomics, we have found that three of each genotype was sufficient to make conclusions in the case of these conditional MR KO mice. Although we did not perform an *a priori* power analysis to determine the numbers of animals required, we found that three animals of each genotype was sufficient given the effect size: most of the 'CA2 genes' were not detected in the knockout (KO), and most of the upregulated 'CA1 genes' are never seen in the CA2 region of the wild type (WT). Nevertheless, we did have data from two more animals (1 female MR KO and 1 female WT) that showed similar results, and so we have now added them to the analysis/included in Extended data Fig. 5. Studies with smaller effect sizes, such as what might be expected with stress or drug treatments, would surely require more animals.

Reviewer #2:

Summary

This study follows on from groundbreaking earlier work to characterise the changes occurring in the CA2 area of the hippocampus following conditional knock out of MR, addressing a key gap in current literature and generating important new knowledge. This work may provide new insights relevant for the study of autism, a syndrome linked to MR gene variants.

Overall impression of the work

The manuscript is clearly written and flows well. Methods used are well justified although generally lacking in detail. The outcomes are precise measurements of gene expression analysis to characterise the neuronal identity of this area of the brain, providing spatial

resolution of changes. The work contributes new insights into the to the research field . Main conclusions are justified.

We thank the reviewer for his/her suggestions and for appreciating the importance of our work. We have now made revisions according to the reviewer suggestions (see below).

Specific comments, with recommendations for addressing each comment

1. Methods should include more detail about animal experiments. The ARRIVE guidelines offer a good framework to adhere to in order to maximise the reproducibility and reuse of the work. Details about the underlying genetic background of all mice used in the different analysis, housing conditions and husbandry should be included. How were mice allocated to each experiment and at what stages were experimenters blinded to genotype? It is not entirely clear if manual spot selection was used for both WT and KO samples, or just for KO samples. Whilst it is understandable why automatic selection was not possible for KO mice, the default should be to apply manual selection based on anatomical location to both sample types by an experimenter blinded to genotype so 'CA2' spot selection was standardised across samples since using different methods could introduce bias. Please provide more detail of how regions (spots) were selected in both types of samples to ensure this is clear. If no blinding/randomisation took place state this explicitly. Whilst studies on female mice have been conducted the data is not presented, why?

In general, we aim to provide detailed methods in all our manuscripts, but we agree that we fell short here in this case, and so we have revised our Methods section accordingly. Specifically, we have now a) provided more details on the genetic backgrounds and husbandry; b) provided more detail on how mice were allocated and blinding protocols; and c) provided more detail on spot selection (all in Methods, page 5). Manual spot selection was based on anatomical location in both cases, but we used CA2 markers in the WT as a guide (this was not possible in the KOs, even using 'CA1 genes'). We agree this would have best been performed by an experimenter blinded to genotype, and we have now explicitly stated that this was not the case; d) we are still acquiring data from sufficient numbers of animals to perform direct comparisons between males and females, which is the topic of a separate study. However, because the androgen receptor (AR) is expressed at comparatively high levels in CA2, and because MRs can be antagonized by progesterone, we had analyzed males and females separately out of caution. However, we have now added data from a pair of female mice (1 WT and 1 MR KO), presented in Extended Data Fig. 5, which show that the changes with MR KO are very similar across all animals. We now also mention this in the Methods, page 5. We also note that both male and female mice were used in the histological experiments, but we did not power the study to detect sex differences.

2. *Please justify n-numbers used for animal studies in text (power calculation including effect size, variance and power or other justification). Also include a statement about the total number of mice used in this manuscript (ST, 6 mice; SMF, 2 mice; Histology, 15 mice = 23?) and detail any ethical approvals for the study (this was ticked in checklist but I could not find this statement in the manuscript).*

As stated above for Reviewer 1, we had not performed a power analysis for the spatial transcriptomics given the very large effect size that was apparent on visual inspection of the gene expression. Nevertheless, we have added the data from additional animals to the data, now presented in Extended Data Fig. 5 (1 each of female WT and MR KO mice), which did not change our conclusions. In addition, we now show in Extended data Fig.5 that the variability between all mice (3 males and 1 female for each genotype) is quite low.

Regarding the numbers for histological study, we have now also included information on the number of mice used for each experiment and noted approval by our Animal Care and Use Committee.

3. What was the inclusion overlap for the DEG analysis i.e. were all genes expressed in any sample included or did you have a stringency cutoff of gene must be expressed in 2/3 samples per group or all 3/3 samples? What was the threshold for the gene to be expressed?

In this study, a gene was considered expressed if its normalized expression value was greater than 0.2 in at least two out of three samples within at least one group (genotype). This information is now included in the Methods.

4. *Fig2. Include name of test used for correlation between groups.*

We are unsure of exactly which test the reviewer is asking about. We had calculated the correlation coefficients between the regions in the WT and MR KO (shown at the bottom of the PCA plots in Fig. 2b). Is the reviewer asking about whether those are significantly different between the WT and KO? We agree that we did not report that comparison directly, but we have now done so. The correlations were tested using Fisher z-transformed correlation values, followed by a standard test in a normal distribution using `pnorm()` in R. These two correlations are significantly different from each other ($p < 0.001$). We hope this is the test in question and are now reported on page 3 and detailed this in the Methods of the revised manuscript.

5. *Extended data 2: More information required relating to this figure. Assignment of cell types to clusters should be provided on the figure or in the legend. Please elaborate on the occurrence of batch effects, I assume from the figure that WT1, 3 and KO1 were analysed separately from WT2, KO2 and KO3. If so, these batch effects seem substantial. In light of such clear batch effects how can the results be relied upon? You state that “Note images of WT3, KO2, and KO3 were flipped horizontally 330 for illustrative purposes.” Is this correct (WT2?) and is it relevant given low-dimensional embedding produced by UMAP are preserved regardless of how the embedding is rotated or reflected? Also there is a 1 missing off the bottom UMAP2 axis (should read 10). Please identify the different wildtype backgrounds be identified in Extended Fig. 2A (Nr3c2 fl/fl and 1 C57B6J wildtype).*

We thank the reviewer for noticing this issue. Indeed, samples from WT1, 3, and KO1 were prepared and sequenced separately from WT2 and KO 2, 3. While we too had noted the apparent processing effects on the hippocampal clusters, we hadn't noticed the large effects in

the structure of UMAPs as pointed out. After digging into the cause, we found that we could better correct for these effects using Harmony. Extended Data Fig. 2 has now been replaced in its entirety, showing all data plotted by cluster, with their locations in the brain samples (in their original orientations; panels a and b) and with spots selected for analysis with their locations within the individual brain sample (panels c, d). Note that the processing effects are still visible in the *stratum radiatum* of CA1 (olive color in panel b). However, these differences were likely due to those areas not yielding enough mRNA for analysis (more evident in panel d), and not that the clustering was fundamentally different in those brains. Our motivation for showing the UMAPs was merely to illustrate that the molecular profile of the brain was largely unchanged in the MR KO and not that the clusters were used for any further analysis. We think our new analyses clarify (and correct) most of the concerns raised.

6. Extended data 5: “Upstream regulators affected by MR KO in CA2 are shown. PRKAA1 and SPRY2 regulators were found to be ‘inactive’ in the regulation of target genes (activation z-score < 0), while other regulators, such as MECP2, were ‘active’ in the regulation of target genes like FKBP5 (activation z-score > 0)”. This interpretation of the upstream regulation doesn’t make sense to me the up stream regulators are not ‘inactive’ the score is worked out on the data from their downstream targets in the data set. The overlap p-value indicates if there is a statistically significant overlap between genes in the dataset and those regulated by the upstream regulator, all of which are highly significant. The z-score indicates the magnitude of the regulation and the sign reflects the predicted activation state, Values <-2 or >+2 are considered significant with a positive number predicting activation of the upstream regulator and a negative value predicting inhibition of the upstream regulator. So based on table b PRKAA1 is predicted to be significantly inhibited as an upstream regulator so its down stream targets will be up or downregulated depending if PRKAA1 affects their expression in a negative or positive way respectively. Whilst the other upstream regulators may be playing a role (due to significant overlap of target genes) the directionality cannot be predicted as z-scores are less than ±2. See Microsoft Word - Upstream Regulator Analysis_Whitepaper_08-22-12_KB.docx (ingenuity.com) for more details.

We thank the reviewer for pointing out this oversight. We have now modified our figure and text to address the statistical significance of these upstream regulators (now Extended data figure 6b).

Reviewer #3:

This is a short manuscript with extensive data that show that in absence of the mineralocorticoid expression, CA2 pyramidal neurons in the mouse hippocampus do not properly differentiate (as shown earlier), but rather acquire a CA1-like phenotype. This is of relevance, given that absence of MR is associated with autism, and CA2 is important for social memory formation. And, it is, or would be, fascinating to learn about out how this receptor for a stress/circadian hormone affects neuronal end-differentiation.

The authors combined spatial transcriptomics with RNAseq, in situ hybridization, and cell nuclear density analysis to make their point. In addition to their main point, the data also are

very informative about the MR-dependent genes in the other hippocampal cell types, something that is overdue – and very good to see.

There are no major issues regarding the approach or main conclusion. Indeed, snRNAseq is not appropriate, given that it would be impossible to trace the ‘former’ CA2 neurons in the mutants.

We thank the reviewer for their kind comments.

As for additional data – it would of course be interesting to know what happens to the projections to the ‘cells formerly known as CA2 neurons’. But I can see that that is certainly a different project.

We have been interested in this very point for some time now. In fact, we have tried to trace CA2 axons using viral labeling techniques, but mostly without success. We think this is because an AAV receptor, KIAA0319L, which is highly expressed in CA2, is not detected in the MR KO tissue (the topic of another manuscript, now submitted, but see an early version on BioRxiv). One serotype, AAV6, does not require this receptor for infection, and so does allow for labeling of CA2 axons (they are apparently decreased in CA1 *stratum oriens*). Unfortunately, because we are not 100% confident that the infection is comparable in the MR KO mice, we did not feel comfortable presenting these data. We are continuing to investigate this topic.

Depending on the plans of the authors, it might be nice to see the pathway enrichment analyses for the other subfields. The data also may contain interesting facts in relation to neurogenesis and apoptosis in the DG, but there I can also see that the framing of the story would become less focused. For one thing, the data on differentially expressed genes in the other subfields should become available.

We thank the reviewer for the suggestion. As this manuscript was originally formatted for a (very) brief communication, we had not presented those data on the other subfields. We are happy to now include it (now shown in Extended Data Figs 7, 8).

One question which may in fact be resolved with immunohistochemistry (or perhaps was addressed in previous work): what of the heterozygous animals? Is half the gene dose sufficient to have the CA2 develop? That may also be relevant in relation to the ASD that develops as a consequence of MR mutations.

We have not studied the MR hets in detail, but we have stained a few sections, which showed that CA2 markers are at least still present in the hets. As this is clearly of interest in the case of the ASD, we are pursuing specifically those mutations as part of a different project, and we will certainly be including heterozygous animals in those studies (perhaps some mutations are dominant-negative or the changes are more subtle than in homozygous animals).

Minor:

1. 'including' – which other corticosteroid receptors are there? (I guess perhaps the AR for DHEA, but nomenclature would say MR and GR are all?).

We thank the reviewer for noting this. We have now changed our wording.